# Divergent Tasks Harm Integration of New Entities via Fine-Tuning

## Abstract

When a pretrained model is fine-tuned to incorporate new entities, does this knowledge automatically generalize across tasks? We investigate this question using a controlled framework in which transformers pretrained on geometric tasks over real-world city coordinates are fine-tuned to integrate synthetic out-of-distribution cities. We find that cross-task generalization varies dramatically depending on the fine-tuning task, and that this variation is partially predicted by representational similarity (CKA) measured during pretraining. In multi-task fine-tuning, we identify lurking *divergent tasks* that not only fail to generalize but actively harm performance on other tasks. Probing suggests that divergent tasks encode new entities in separate subspaces rather than integrating them into the shared world manifold.

## 1 Introduction

Neural networks can develop structured internal representations of their training data (Bengio et al., 2014; Li et al., 2022; Gurnee & Tegmark, 2023), but major questions remain about how these representations adjust during fine-tuning. In real-world settings, key factors (world, data, model) are entangled and costly to vary independently. We develop a synthetic framework where these factors can be precisely controlled.

**This work.**    To study these questions, we decouple the underlying *world* from the *data generation process* to control them independently. Concretely, we adopt the coordinates of real-world cities as our "world," a ready-made complex structure with ground-truth geometry, and define 7 geometric tasks on top of it. We train autoregressive Transformers on this data, then test whether models can integrate new entities (`Atlantis` cities) into their learned representations via fine-tuning. Our main contribution is:

- **A Framework Decoupling World, Data and Model.** We separate the underlying world (city coordinates) from the data generation process (7 geometric tasks), enabling systematic study of how different tasks shape representations of the same world. The world provides ground-truth coordinates for directly assessing representation quality via probing. This setup also allows defining consistent world updates (adding synthetic `Atlantis` cities) to test whether models can adapt their representations accordingly.

- **Divergent Tasks Harm Fine-Tuning of New Entities Despite Multi-Task Pretraining.** We test whether models can integrate new entities (`Atlantis` cities) via fine-tuning. We find that single-task representational similarity (CKA) partially predicts cross-task generalization. In a multi-task fine-tuning setting, we find surprising "divergent" tasks which hinder integration of new entities into the learned manifold, actively harming generalization.

For related work, see App. A.

## 2 Results

We train small transformers on 7 geometric tasks over 5,075 real-world city coordinates (Fig. 1). Our main model is trained on all 7 tasks jointly; this multi-task pretraining yields representations

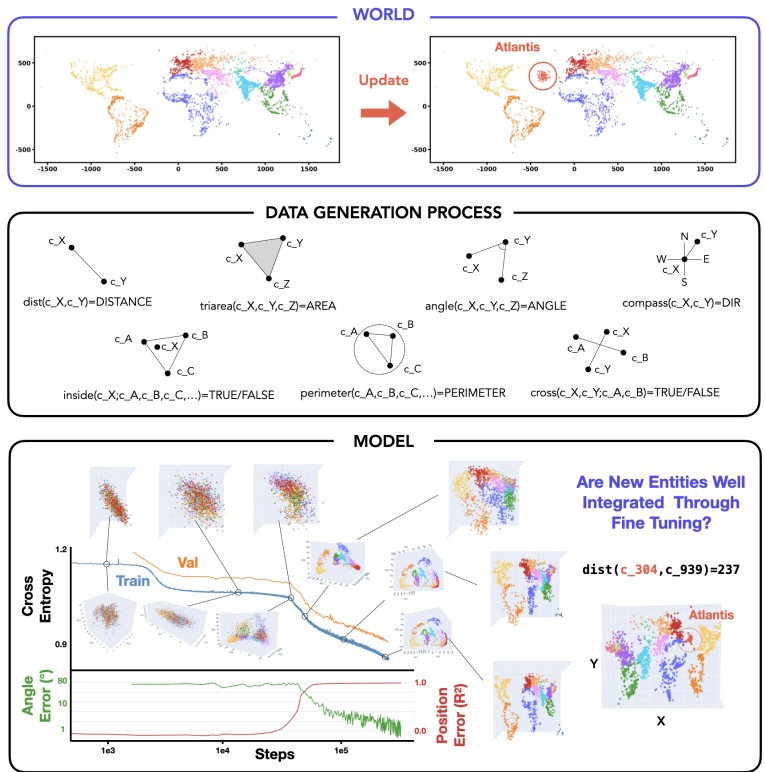

Figure 1: **Overview of the World-Data-Model framework. Top:** The world consists of 5,075 real city coordinates; we test adaptation by adding 100 synthetic `Atlantis` cities (App. C.1). **Middle:** Seven geometric tasks generate training data from city coordinates (App. C.2). **Bottom:** Training dynamics of one model, showing loss curves, linear probing accuracy for coordinate reconstruction and visualizations of internal representations (PCA and linear probe projections) at different training stages. See App. Fig. 6 for all training curves.

where PCA naturally reveals world map structure (Fig. 2). We also train 7 single-task models (one per task) and measure their representational similarity using CKA (Kornblith et al., 2019), which we call *single-task CKA*. See App. C for full details.

To test adaptation, we introduce 100 synthetic `Atlantis` cities (Atlantic Ocean) never seen during pretraining. We fine-tune the 7-task model to integrate `Atlantis` and test whether this generalizes across tasks. If representations are properly factored, `Atlantis` should integrate seamlessly; if not, we suspect either fractured representations (Kumar et al., 2025) or gradient descent failing to trigger proper updates (Kumar et al., 2022).

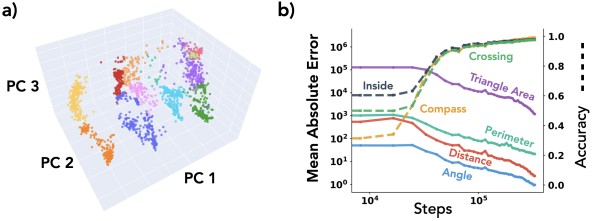

Figure 2: **7-task model.** (a) PCA projection of layer 5 representations naturally reveals world map structure. (b) Training curves showing successful learning of all 7 tasks, including `crossing` which failed in single-task training.

**Result 1: Pretraining Phase Representational Alignment Predicts Fine-Tuning Generalization *Despite* Joint Pretraining** We first address a simple question: when fine-tuning on `Atlantis` cities for a single task (e.g., `distance`), should we expect the model to automatically generalize to using `Atlantis` for all other tasks?

To answer this, we fine-tune on 100k examples of a single task that include `Atlantis` cities, mixed with original pretraining data to avoid catastrophic forgetting and a small multi-task elicitation set (see App. C.3 for details).

The resulting generalization matrix is shown in Fig. 3(a). This matrix reveals rich phenomenology: some tasks like `distance` show no cross-task generalization (`Atlantis` remains usable only for that task), while `angle` triggers significant generalization across all tasks. Intriguingly, we observe an apparent inverse relationship: tasks that efficiently trigger cross-task generalization of new entities are often those that don't easily benefit from other tasks' fine-tuning, though this relationship is noisy.

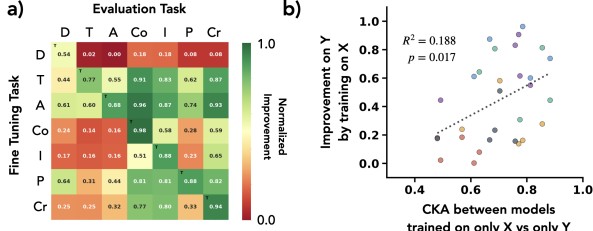

Figure 3: **Fine-tuning generalization correlates with single-task CKA.** (a) Generalization matrix: rows = fine-tuning task, columns = evaluation task, values = normalized improvement on `Atlantis` (App. D.1). Averaged over 4 seeds. (b) Single-task CKA vs. normalized improvement (App. Fig. 13 for labels).

Unexpectedly, we find that *generalization performance correlates with the CKA values from single-task pretraining*. This is puzzling: the CKA values come from models trained from scratch on individual tasks, yet they partially predict fine-tuning behavior of a model pretrained on all tasks jointly (Fig. 3b). If the multi-task model truly uses unified representations for cities, why would single-task representational properties matter?

For clarity, we define two terms: **Divergent tasks** are tasks which have low CKA compared to others when trained in isolation (in our case the `distance` task). **Hidden spaces** are representation spaces not surfaced by PCA or probing but used by divergent tasks.

We hypothesize:

> *"Even though models develop joint world representations which converge in multi-task pretraining, gradient descent on divergent tasks might fail to act on these shared representations during fine-tuning, instead utilizing hidden spaces that don't propagate updates across tasks."*

Our question is then two-part:

1. To what extent do divergent tasks affect fine-tuning generalization?
2. Will gradient descent on divergent tasks fail to merge fine-tuning introduced concepts to the original representation manifold?

**Result 2: Divergent Tasks Catastrophically Harm Generalization** To investigate how divergent tasks affect generalization, we move from single-task to multi-task fine-tuning settings. First, we introduce a simple heuristic model: fine-tuning on a concatenated dataset $\{D_1, D_2, ..., D_n\}$ (which do not provide conflicting supervision) should combine their individual effects. Specifically, when concatenating and shuffling all fine-tuning data to avoid sequential learning effects like catastrophic forgetting (McCloskey & Cohen, 1989), we expect the improvement $\mathrm{Imp}_i$ on task $i$ after training on tasks $j$ and $k$ to follow a **best-teacher model**:

$$\mathrm{Imp}_i(D_j \cup D_k) = \max(\mathrm{Imp}_i(D_j), \mathrm{Imp}_i(D_k)) \tag{1}$$

To test this hypothesis, we fine-tuned the 7-task model on all $\binom{7}{2} = 21$ possible two-task combinations. Fig. 4(a,c) shows the *deviation* from our best-teacher expectation (averaged over 4 seeds; see App. Fig. 15 for raw improvements and App. Fig. 16 for individual seeds). Strikingly, we observe "red horizontal bands", models that not only fail to benefit from multi-task training but actually perform worse than their best single-task component. Notably, all these degraded performance bands involve the `distance` task. Fig. 4(c) quantifies this: when we split the deviation values into models with and without `distance`, we consistently observe lower-than-expected performance when the divergent task is included. This confirms that *divergent tasks (those with low single-task CKA)*

*actively harm fine-tuning generalization rather than simply failing to contribute.* We next examine how this manifests in the learned representations.

**Result 3: Divergent Tasks Disrupt Representational Integration of New Entities**  Having shown that divergent tasks harm generalization (Question 1), we now address Question 2: does gradient descent on divergent tasks fail to merge new entities into the representation manifold?

We take two exemplars from the 21 fine-tuning runs: one without `distance` that generalized well (`angle` + `compass`), and one with `distance` that was harmed (`distance` + `perimeter`). We first train a linear probe on a subset of all cities including `Atlantis`; these reconstructions are shown in Fig. 4(b) (top and bottom panels). In the well-integrated case, `Atlantis` cities lie within the world data manifold. In the ill-integrated case, `Atlantis` cities are off the manifold. While this difference appears subtle in 2D projections, the effect is dramatic in

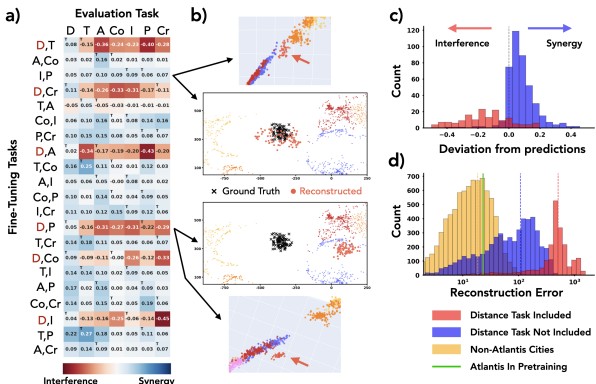

Figure 4: **Divergent tasks harm multi-task fine-tuning and disrupt representational integration.** (a) Deviation from best-teacher expectation for 21 two-task models (rows) across 7 evaluation tasks (columns); "red horizontal bands" show `distance` combinations degrade performance. (b) Representation visualization and linear probe reconstruction of `Atlantis`. (c) Histogram of deviation values. (d) Linear probe reconstruction error; green line = `Atlantis` in pretraining. 3D visualizations: link.

3D—we strongly encourage readers to explore our interactive visualizations . Next, we train a linear probe on 4000 non-`Atlantis` cities and apply it to `Atlantis` representations (middle panels). In the well-integrated case, `Atlantis` cities (red-orange) are relatively well reconstructed compared to ground truth (black crosses); in the ill-integrated case, reconstruction fails completely.

We quantify this effect in Fig. 4(d), showing histograms of absolute coordinate reconstruction error. When `Atlantis` is integrated via fine-tuning partially on divergent task data (red), reconstruction errors are nearly an order of magnitude larger than when integrated via purely non-divergent tasks (blue). For reference, non-`Atlantis` cities (yellow, still held out from probe training) show low reconstruction error as expected. One might hypothesize that `Atlantis`'s location in the middle of the ocean creates inherently difficult geometry. To test this, we pretrained a model with `Atlantis` included from the start (green line). In this case, `Atlantis` cities are reconstructed as well as any other city, confirming that the integration failure stems from divergent task fine-tuning dynamics rather than geographic peculiarity.

**This suggests that divergent tasks cause optimization to encode new entities in hidden spaces rather than integrating them into the existing world manifold, explaining their failure to support cross-task generalization.**

Putting these results together: single-task representational divergence weakly predicts fine-tuning generalization even after joint pretraining, and the most divergent task (distance) actively harms integration of new entities. This raises a hypothesis: certain task-architecture pairings may have intrinsic properties that induce gradient dynamics bypassing shared representations, causing updates in hidden subspaces that harm generalization, even when the network uses unified representations for the forward pass.

**Limitations.**  Our findings are correlational: we do not claim that interventions to increase single-task CKA would necessarily improve fine-tuning generalization. Rather, we identify representational divergence as a diagnostic marker for tasks that will harm multi-task fine-tuning performance.

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

# APPENDIX

## A RELATED WORK

**Internal Representations.** Understanding internal representations has roots in neuroscience (Hubel & Wiesel, 1962), informing early neural network development (Fukushima, 1980; Bengio et al., 2014; Rosenblatt, 1958; Rumelhart et al., 1986). Recent work has revealed that language models develop structured "world models" encoding geographic, temporal and relational information (Li et al., 2022; Gurnee & Tegmark, 2023; Nanda et al., 2023; Marks & Tegmark, 2024), with similar representations emerging during in-context learning (Vafa et al., 2025). Mechanistic interpretability and sparse autoencoders have enabled decomposition of neural activations into interpretable features (Anthropic AI, 2023; Templeton et al., 2024). Researchers have also uncovered that models represent meaningful properties of data, including concepts (Pearce et al., 2025; Higgins et al., 2017), features (Olah et al., 2017), and abstractions (Lee et al., 2025; Arditi et al., 2024), in interpretable ways. Furthermore, PRH posits that diverse models converge toward similar representational structures (Huh et al., 2024). However, recent work questions this representational optimism, suggesting that deep network representations may be more brittle than previously assumed (Kumar et al., 2025). Only recent work has begun examining how representations emerge during pretraining in real LLMs (Li et al., 2025; Ge et al., 2025) or how they change during fine-tuning (Lee et al., 2024). Our work takes a complementary perspective, studying the factors that control the formation of these representations and how networks integrate new entities into their representation space via fine-tuning.

**Fine-tuning.** The pretraining-finetuning paradigm has become central to modern deep learning, with seminal works establishing its effectiveness in computer vision (Krizhevsky et al., 2012; He et al., 2015) and natural language processing (Devlin et al., 2018; Radford et al., 2018). Despite widespread success, fine-tuning exhibits poorly understood behaviors such as the reversal curse (Berglund et al., 2024; Lampinen et al., 2025), out-of-context reasoning limitations (Treutlein et al., 2024), and off-target effects (Betley et al., 2025). On this background, careful studies of fine-tuning and other low-compute adaptation methods have raised pessimism about whether models can learn fundamentally new abilities, suggesting they may merely form "thin wrappers" around pretrained representations (Jain et al., 2023; Ward et al., 2025; Yue et al., 2025; Qin et al., 2025; Zhao et al., 2025; Zweiger et al., 2025). Fine-tuning has also been studied across diverse directions: parameter efficiency (Hu et al., 2021; Lester et al., 2021), zeroth-order optimization (Malladi et al., 2024), weight composition (Ilharco et al., 2023), and representation adaptation (Wu et al., 2024). Work on feature distortion (Kumar et al., 2022) is perhaps most related to ours, though representational changes are assumed rather than directly measured. Our work examines this question in a controlled setup where ground-truth world structure enables precise measurement of representation adaptation.

**Dynamics of Representations.** Recent work has begun studying how representations evolve during in-context learning (Shai et al., 2025; Demircan et al., 2024) or fine-tuning (Casademunt et al., 2025; Minder et al., 2025). Relatedly, Lubana et al. (2025) show that representations exhibit rich temporal dynamics that standard interpretability methods (e.g., SAEs) fail to capture due to stationarity assumptions. Fu et al. (2025) show that VLMs trained by merging LLMs and vision encoders often fail to utilize representations surfaced by the vision encoder, i.e. the representations exist but remain unused.

**Geometric Deep Learning.** Geometric deep learning studies how data geometry interacts with model architectures, developing equivariant networks that respect symmetries (Bronstein et al., 2021; Cohen & Welling, 2016; Weiler & Cesa, 2021). While our world is defined on a 2D plane, one might ask: why not a sphere, torus, or other manifold? This is an interesting direction, but not our focus. We study how neural networks adapt internal representations to tasks in an arbitrarily chosen geometry. Moreover, a change in world geometry can be absorbed into the task definition (e.g., geodesic vs. Euclidean distance), so the key question remains how representations form given the task, not the underlying manifold. Planar coordinates also allow clean linear probing of world representations. Our models are standard transformers without geometric priors; we study what representations emerge purely from training on task data, treating geometry as emergent rather than imposed.

**Loss Plateaus.** Our `crossing` task fails to learn in single-task training despite escaping an initial plateau (likely output format learning), suggesting it remains stuck in a deeper plateau. Such plateaus are notoriously difficult for transformers. Recent work has studied this phenomenon mechanistically in transformers (Hoffmann et al., 2024; Gopalani & Hu, 2025; Singh et al., 2024), while others relate it to more general optimization challenges in deep learning such as simplicity bias and gradient starvation (Shah et al., 2020; Pezeshki et al., 2021; Bachmann & Nagarajan, 2025). Most related to our findings, Kim et al. (2025) show that multi-task training shortens loss plateaus, similar to why our `crossing` task trains successfully when joined with any other task.

## B  3D VISUALIZATIONS

3D visualizations are available here (Open Science Framework link).

## C  EXPERIMENTAL DETAILS

This section provides detailed information about the world, data generation process, model architecture, and training procedures used in our experiments.

### C.1  WORLD

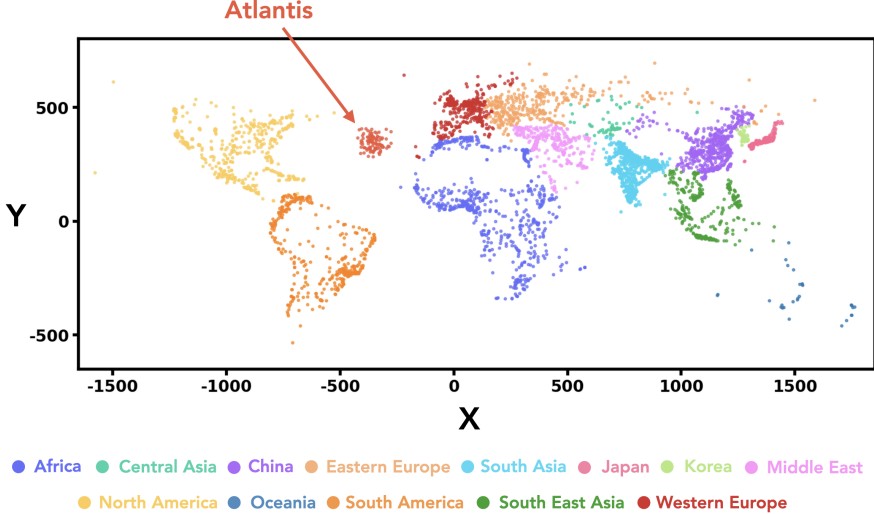

Figure 5: **Geographic distribution of cities used in our experiments.** 5,075 real-world cities plus 100 synthetic `Atlantis` cities (5,175 total). Cities span all continents and provide a fixed, measurable world structure. Coordinates use an equirectangular projection: $x = 10 \times$ longitude, $y = 10 \times$ latitude (in degrees). The `Atlantis` region (Atlantic Ocean) is used for out-of-distribution testing.

Our experiments use a geographic world consisting of 5,075 cities extracted from the GeoNames (OpenDataSoft / GeoNames, 2025) database with population greater than 100,000. Cities are distributed across all continents. This choice provides natural variation in density (e.g., dense regions like India versus sparse Oceania) that creates interesting computational challenges.

While we use real city coordinates, this work studies abstract geometric reasoning rather than actual geography—we project coordinates to Euclidean space using an equirectangular projection (as described above) and treat all tasks as pure geometry problems.

We deliberately chose a flat 2D manifold rather than a spherical globe. Our early experiments used spherical coordinates, but we realized that regardless of the external world's geometry, the model must construct its own internal representation starting from random entity distributions. Given the

540 model's nonlinearity, there is no fundamental reason why any particular geometry (planar, spheri-
541 cal, etc.) would be canonical. Our choice of planar geometry enables clean linear probing to read
542 out world representations, whereas extracting nonlinear manifold structure remains an open chal-
543 lenge (Engels et al., 2024; Csordás et al., 2024). While geometric deep learning (Bronstein et al.,
544 2021) studies the interaction between data geometry and model computation, our focus is on general
545 sequence modeling rather than geometry-aware architectures.

546 Additionally, we introduce 100 synthetic `Atlantis` cities positioned in the Atlantic Ocean, cen-
547 tered at (longitude $-35°$, latitude $35°$) and following a Gaussian distribution with standard devia-
548 tion of $3°$. These synthetic cities enable controlled out-of-distribution experiments, as models never
549 observe `Atlantis` during pretraining but must generalize to it during evaluation. City IDs are ran-
550 domly assigned from the range [0, 9999], creating a sparse identifier space that models must learn
551 to map to coordinates. All coordinates are stored as integers (after the $\times 10$ scaling), eliminating
552 floating-point precision issues.

## C.2 DATA GENERATION PROCESS

**Tasks** We implement 7 geometric tasks that operate on city coordinates. All tasks use a consistent
format: `task(arguments)=answer`, where city IDs are prefixed with `c_`. Numerical outputs
(distance, area, angle, perimeter) are rounded to integers. Table 1 summarizes the tasks.

| Task | Input | Output Type | Unit/Values | Example |
|---|---|---|---|---|
| distance | 2 cities | Numerical | Scaled coords | `dist(c_865,c_4879)=769` |
| triarea | 3 cities | Numerical | Scaled coords$^2$ | `triarea(c_1234,c_5678,c_9012)=45823` |
| angle | 3 cities | Numerical | Degrees (0–180) | `angle(c_2345,c_6789,c_123)=97` |
| compass | 2 cities | Categorical | 8 directions | `compass(c_1234,c_5678)=NE` |
| inside | $1+n$ cities | Categorical | TRUE/FALSE | `inside(c_9012;c_3456,...)=FALSE` |
| perimeter | $n$ cities | Numerical | Scaled coords | `perimeter(c_4567,c_8901,...)=2856` |
| crossing | 4 cities | Categorical | TRUE/FALSE | `cross(c_2345,c_6789;c_123,c_4567)=TRUE` |

Table 1: Summary of 7 geometric tasks. Numerical outputs are integers; "scaled coords" refers to
the $\times 10$ coordinate system (Sec. C.1). Categorical tasks have discrete outputs: `compass` uses 8
cardinal directions (N, NE, E, SE, S, SW, W, NW), while `inside` and `crossing` are binary. The
`inside` task tests if the first city lies within the convex hull of the remaining cities; `crossing`
tests if line segment $(c_1, c_2)$ intersects segment $(c_3, c_4)$.

It is important to note that for all tasks we study, queries that don't explicitly involve `Atlantis`
cities maintain identical outputs after `Atlantis` is introduced—ensuring we can cleanly measure
integration of new knowledge. While our framework could be extended to study tasks where exist-
ing answers change (e.g., counting cities within a radius would yield different results after adding
`Atlantis`), enabling investigation of phenomena like the reversal curse (Berglund et al., 2024),
we focus here on the simpler case of integrating new entities while preserving existing knowledge.

**Dataset Sizes** Each pretraining set consists of 1M rows of data per task. For fine-tuning, the
dataset consists of: (1) 100k rows of the target task containing at least one `Atlantis` city, (2)
20k rows randomly sampled from the original pretraining data to prevent catastrophic forgetting,
and (3) 256 rows per task (without `Atlantis`) to elicit multi-task performance. For the baseline
experiment where `Atlantis` is included during pretraining (green line in Fig. 4d), we use 1M
rows per task but sample cities uniformly without treating `Atlantis` specially.

## C.3 MODEL AND TRAINING

**Tokenization** We use character-level tokenization with 98 ASCII tokens (excluding space, which
serves as the delimiter), plus special tokens for beginning-of-sequence (BOS), end-of-sequence
(EOS), and padding (PAD). Each task query and answer is tokenized character-by-character
(e.g., `dist(c_0865,c_4879)=769` becomes `d i s t ( c _ 0 8 6 5 , c _ 4 8 7
9 ) = 7 6 9`).

This character-level scheme is intentional. While assigning each city and task a dedicated token
would simplify learning, such synthetic-friendly tokenization does not reflect how real language

models operate. LLMs must handle multi-token entities, variable-length prompts (our task prefixes have different lengths), computations at different sequence positions, and irregularly tokenized content (e.g., numbers in LaTeX). Preliminary experiments exploring pitfalls of next-token prediction (Bachmann & Nagarajan, 2025) showed that tokenization details qualitatively affect results. We therefore chose character-level tokenization to better approximate realistic sequence modeling conditions.

**City ID Assignment** City IDs are randomly assigned from the range $[0, 9999]$, ensuring no geographic information leaks through the identifier. This random assignment means the model cannot exploit ID patterns to infer coordinates.

**Architecture** We use the Qwen2 (Yang et al., 2024) decoder-only transformer architecture with hidden size 128, 4 attention heads, and 6 layers.

**Pretraining** We train models autoregressively on the full sequence (no prompt masking). While we observed training speedup when masking loss computation on the prompt side, we deliberately avoid this optimization to maintain similarity with standard autoregressive language model pretraining. All pretraining runs see 42M rows regardless of dataset size (e.g., 42 epochs for 1M rows, 6 epochs for 7M rows). Table 2 summarizes the hyperparameters.

| Hyperparameter | Value |
|---|---|
| Optimizer | AdamW (Loshchilov & Hutter, 2019) |
| Learning rate | $3 \times 10^{-4}$ |
| Weight decay | 0.01 |
| Scheduler | Linear with warmup |
| Warmup steps | 50 |
| Batch size | 128 |
| Max sequence length | 256 |
| Total training rows | 42M |
| Initialization scale | 0.1 (std) |

Table 2: **Pretraining hyperparameters.**

**Fine-Tuning** Fine-tuning starts from the final pretrained checkpoint. We use a reduced learning rate of $1 \times 10^{-5}$ ($30\times$ smaller than pretraining) to avoid catastrophic forgetting. The fine-tuning dataset consists of 100k rows per task containing at least one `Atlantis` city. We train for 30 epochs with batch size 128. We observed significant degradation in performance for both the fine-tuned task and original (non-`Atlantis`) tasks when using a larger batch size of 512. All other hyperparameters (optimizer, weight decay, scheduler, warmup) remain the same as pretraining.

# D ANALYSIS METHODS

## D.1 EVALUATION

**Generation Protocol** For evaluation, we use teacher forcing up to the "=" sign (the prompt), then generate autoregressively at temperature zero until reaching the EOS token or a maximum of 128 tokens (sufficient for all tasks). All trained models achieve perfect parse accuracy—outputs always match the expected format (integers for numerical tasks, valid categories for categorical tasks).

**Task-Specific Metrics** Categorical tasks (`compass`, `inside`, `crossing`) are evaluated using accuracy. Numerical tasks are evaluated using absolute error: `distance` (scaled coordinate units), `triarea` (scaled coordinate units$^2$), `angle` (degrees), and `perimeter` (scaled coordinate units).

**Normalized Improvement** To compare generalization across tasks with different metrics and scales, we define a normalized improvement score that maps performance to $[0, 1]$, where 0 indicates no improvement over the `Atlantis` baseline (before fine-tuning) and 1 indicates matching the pretrained model's performance on standard cities.

For **error-based tasks** (`distance`, `triarea`, `angle`, `perimeter`), where lower is better:

$$\text{NI} = \frac{\log(\text{baseline}_{\text{atlantis}}/\text{error})}{\log(\text{baseline}_{\text{atlantis}}/\text{baseline}_{\text{standard}})} \tag{2}$$

The logarithmic scaling ensures multiplicative improvements are treated equally (e.g., reducing error from 1000 to 100 is weighted the same as 100 to 10).

For **accuracy-based tasks** (`compass`, `inside`, `crossing`), where higher is better:

$$\text{NI} = \frac{\text{accuracy} - \text{baseline}_{\text{atlantis}}}{\text{baseline}_{\text{standard}} - \text{baseline}_{\text{atlantis}}} \tag{3}$$

Note that normalized improvement can slightly exceed 1.0 if, by chance, `Atlantis` cities perform better than the average pretrained city on some task.

## D.2 REPRESENTATION EXTRACTION

We extract representations from the residual stream after transformer blocks, specifically at layers 3, 4, 5, and 6 of our 6-layer model. Unless otherwise specified, all representation analyses in this paper use layer 5 representations.

To extract city representations, we pass a task prefix followed by a city ID through the model. For single-task models, we use the corresponding task prefix. For multi-task models (2-task and 3-task), we use the first task in the combination as the prefix. We verified that the choice of task prefix has negligible effect on the extracted city representations.

For a city with ID 1234, the input sequence is:

<bos> d i s t ( c _ 1 2 3 4 ,

We extract and concatenate the representations of two tokens: (1) the last digit of the city ID and (2) the following delimiter token (typically a comma). This yields a 256-dimensional representation ($128 \times 2$) per city, which we use for both PCA visualization and linear probing.

**Omitting cities with leading zeros** We omit cities with IDs starting with 0, 00, or 000 from representation analyses. These cities form distinct clusters in representation space, separate from cities with IDs starting with non-zero digits. We hypothesize this occurs because the digit 0 has special semantic status: in numerical outputs (distances, angles, areas), leading zeros never appear (e.g., "=769" not "=0769"), so the model learns to treat 0 differently when it appears as a leading digit. When 0 appears at the start of a city ID, the model may encode a feature indicating "this is an identifier, not a number," causing these cities to cluster separately. To ensure consistent evaluation across all cities, we exclude IDs matching the pattern `^[0][0-9]*$` (i.e., any ID starting with zero).

## D.3 LINEAR PROBING & PCA

We use the representations described in Sec. D.2 for both PCA visualization and linear probing.

**Linear Probing** We train linear probes to predict city coordinates $(x, y)$ from the 256-dimensional representations. We use a train/test split of 3250/1250 cities, training separate probes for $x$ and $y$ coordinates via ordinary least squares (OLS) without regularization. We report $R^2$ scores and mean absolute error in scaled coordinate units.

**PCA** For visualization, we apply PCA to the representations and plot the first two or three principal components. We use consistent color coding based on geographic region to enable visual comparison across models and seeds.

**Reconstruction Error** To quantify how well new entities (`Atlantis` cities) are integrated into the learned manifold, we train linear probes exclusively on non-`Atlantis` cities and evaluate reconstruction error on held-out `Atlantis` representations. Reconstruction error is measured as the absolute Euclidean distance between predicted and true coordinates. Large reconstruction errors indicate that new entities are encoded in different subspaces than the original cities.

## D.4 CENTERED KERNEL ALIGNMENT

We use Centered Kernel Alignment (CKA) (Kornblith et al., 2019) to measure representational similarity between models. Given two representation matrices $X \in \mathbb{R}^{n \times d_1}$ and $Y \in \mathbb{R}^{n \times d_2}$ (same $n$ cities, potentially different dimensions), we compute linear kernel matrices $K = XX^T$ and $L = YY^T$, center them, and compute:

$$\text{CKA}(X,Y) = \frac{\langle K, L \rangle_F}{\|K\|_F \|L\|_F} \tag{4}$$

where $\langle \cdot, \cdot \rangle_F$ denotes the Frobenius inner product. CKA yields a similarity score in $[0, 1]$ that is invariant to orthogonal transformations and isotropic scaling.

For each pair of models, we extract city representations (Sec. D.2) and compute CKA between the resulting matrices. We filter cities to exclude `Atlantis` and IDs starting with zeros. We report CKA values at layers 3, 4, 5, and 6, with layer 5 as the default unless otherwise specified.

# E ADDITIONAL EXPERIMENTS & RESULTS

## E.1 TRAINING DYNAMICS

Fig. 6 shows training dynamics for all seven single-task models. Each panel displays three rows of metrics over gradient steps: (top) training and validation loss, (middle) task-specific performance metric alongside linear probe $R^2$ for coordinate decoding, and (bottom) linear probing distance error measuring how accurately city coordinates can be reconstructed from representations.

Several patterns emerge across tasks. First, all tasks except `crossing` eventually achieve high coordinate $R^2$ (red curves reaching $\sim$1.0), indicating that world representations form reliably across diverse geometric objectives. Second, the relationship between loss, task performance, and coordinate decodability varies across tasks. Third, `crossing` (panel g) fails entirely in single-task training. Loss remains high, accuracy stays near chance, and coordinate $R^2$ never rises, consistent with the main text observation that this task requires multi-task scaffolding.

**Representation Dynamics.** Fig. 7 visualizes how internal representations evolve during training via PCA projections at six checkpoints. A striking pattern emerges: once a representational structure forms, it remains largely fixed throughout the subsequent training phase where task accuracy continues to improve. Examining the gradient steps, representations are essentially fixed in the first $\sim$15% of training, remaining static while loss slowly decreases and accuracy rises. The `distance` task (top row) establishes its thread-like structure early; `angle` (middle row) settles into a 2D manifold; `compass` (bottom row) forms fragmented regional clusters, all within the first few checkpoints, with minimal subsequent change. What determines when representations stop evolving remains unclear, though it appears correlated with the initial loss drop. This may relate to recently observed gradient dynamics in language model training, where loss deceleration phases exhibit qualitatively different learning behavior (Mircea et al., 2025).

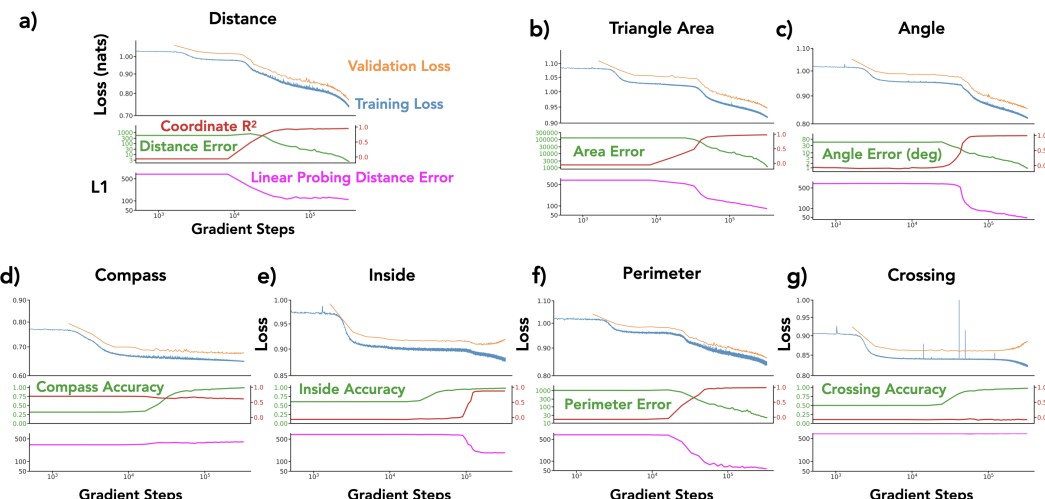

Figure 6: **Training dynamics for all single-task models.** (a) distance, (b) trianglearea, (c) angle, (d) compass, (e) inside, (f) perimeter, (g) crossing. Each panel shows three rows: (top) training loss (blue) and validation loss (orange); (middle) task-specific metric (green, left axis) and linear probe coordinate $R^2$ (red, right axis); (bottom) linear probing distance error (magenta). All plots use log-scale x-axis for gradient steps.

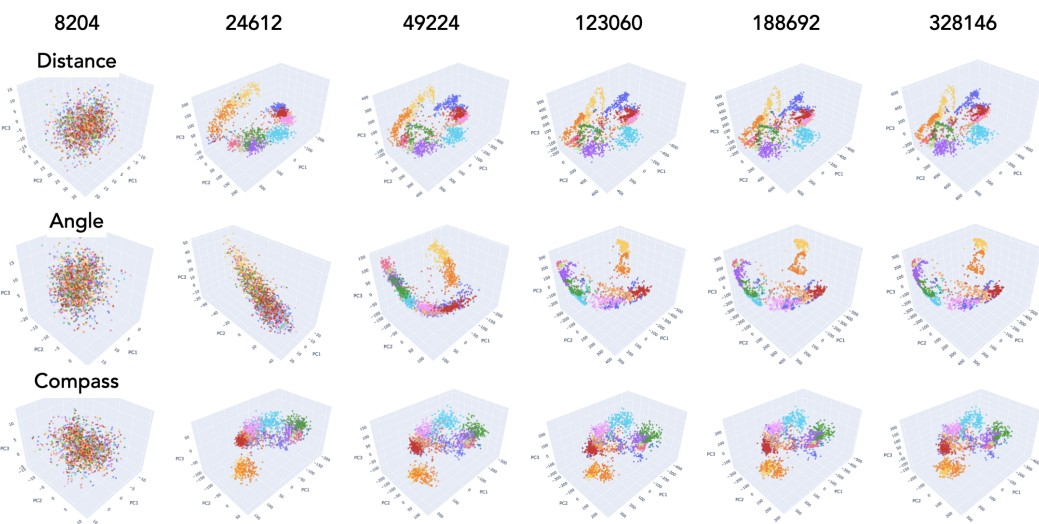

Figure 7: **Representation dynamics during training.** Rows: distance (top), angle (middle), compass (bottom). Columns show PCA projections at gradient steps 8204, 24612, 49224, 123060, 188692, and 328146 (left to right). Cities are colored by geographic region.

### E.2 QUALITATIVE REPRESENTATIONS

Fig. 8 shows PCA projections of city representations for single-task models across three random seeds (rows). The `distance` task consistently produces characteristic thread-like structures. `Angle` and `perimeter` often form larger 2D manifold-like structures. `triangle area` tends to produce arc-shaped geometries. `Compass` forms local clusters corresponding to directional categories, while `inside` produces a more global, diffuse structure.

While there is some seed-to-seed variability within each task, the broader categories remain distinguishable: `distance` representations are qualitatively distinct from the cluster-based representations of `compass` and `inside`, and both differ from the manifold-like structures produced by `triangle area`, `angle`, and `perimeter`.

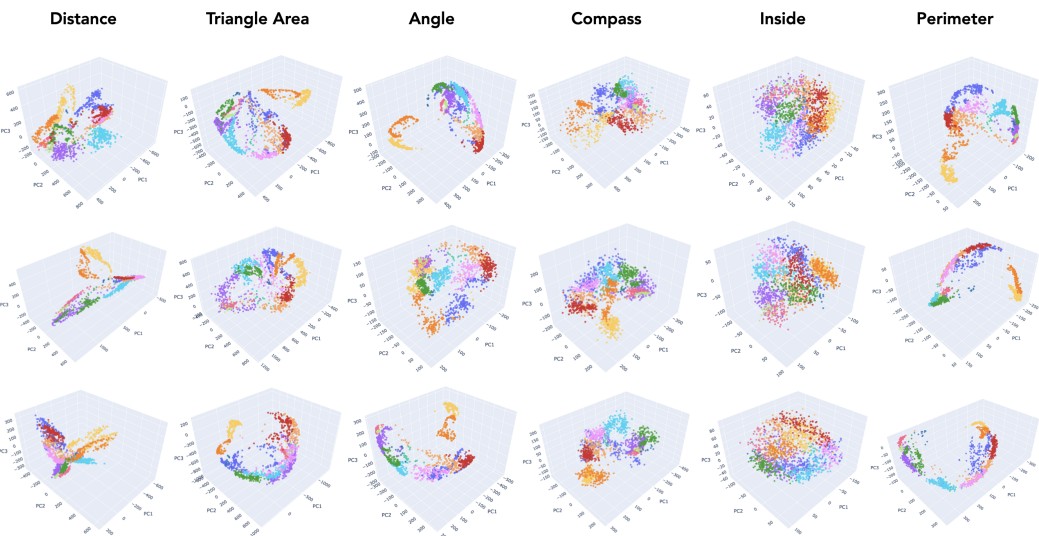

Figure 8: **Representation visualizations for single-task models across multiple seeds.** Each column shows a different task; each row shows a different random seed. Cities are colored by geographic region. Despite seed variability, task-specific geometric patterns are visible.

### E.3 ADDITIONAL CKA RESULTS

**Single-Task CKA Across Layers.** Fig. 9 shows CKA matrices for single-task models at layers 3, 4, 5, and 6. Each cell shows mean ± SEM across 3 seeds. We observe: (1) CKA values increase from layer 3 to layers 4–6, indicating that world representations become more consistent in later layers; (2) the `distance` task (D) shows lower CKA with other tasks across all layers, consistent with its divergent representational geometry; (3) `crossing` (Cr) shows near-zero CKA due to training failure in single-task settings; (4) diagonal entries (same task) can show significant variability, indicating that even identical training objectives can yield different representational solutions.

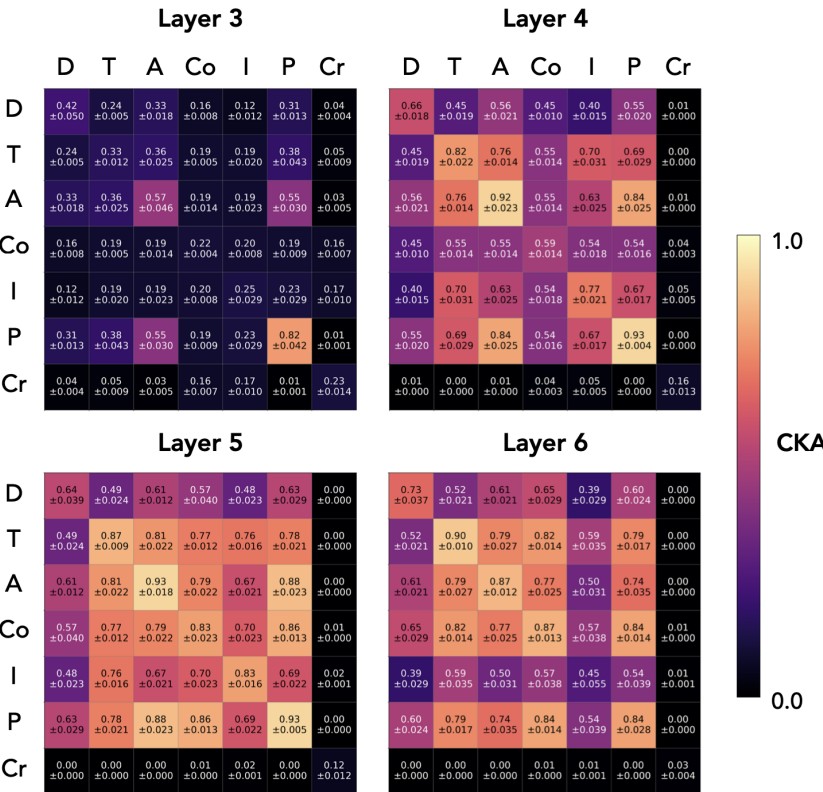

Figure 9: **CKA matrices for single-task models across layers.** Each cell shows mean ± SEM across 3 seeds. D=distance, T=triangle area, A=angle, Co=compass, I=inside, P=perimeter, Cr=crossing. CKA increases in later layers; `distance` shows consistently lower cross-task similarity.

**Two-Task CKA.** Fig. 10 shows the CKA matrix for two-task models at layer 5. Compared to single-task models (Fig. 9, layer 5), two-task training substantially increases representational alignment: all off-diagonal entries exceed 0.84, compared to values as low as 0.48 for single-task models. Notably, diagonal entries (same task combination, different seeds) show minimum CKA of 0.89, indicating that multi-task training also reduces inter-seed variance. For diagonal entries, we exclude same-seed comparisons (which trivially yield 1.0) and report only the upper triangle since the matrix is symmetric. This confirms the main text finding that multi-task training drives representational convergence.

**CKA vs. Task Count (Per-Seed).** Fig. 11 shows the same CKA vs. task count analysis as Fig. **??**(d) in the main text, but broken down by individual seeds. Each panel shows one seed. These per-seed values are pooled to produce the main text figure, where error bars represent SEM across seeds. The pattern is consistent across all three seeds: CKA increases substantially from 1 to 2 tasks and saturates at 2–3 tasks for layers 4–6.

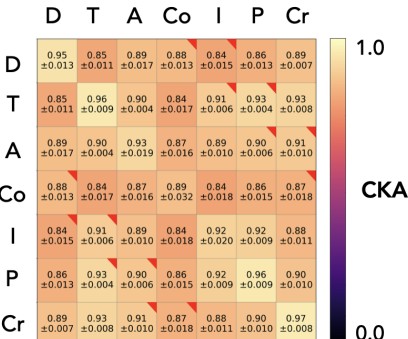

Figure 10: **CKA matrix for two-task models at layer 5.** Mean $\pm$ SEM across 3 seeds. All pairs show high alignment ($>0.84$), substantially higher than single-task models.

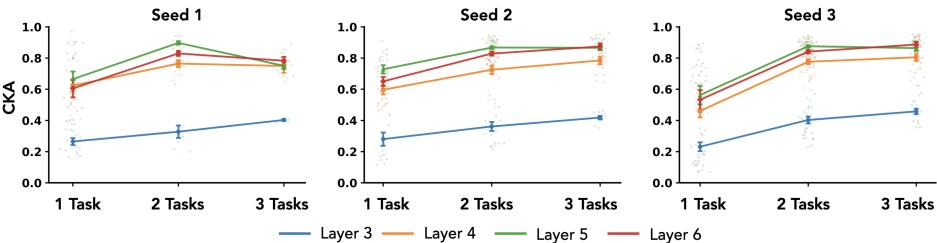

Figure 11: **CKA vs. task count for individual seeds.** Each panel shows a different seed. These values are pooled in Fig. **??**(d); error bars there represent SEM across seeds.

**Aggregated CKA Trends.** Fig. 12(a) shows CKA vs. task count for a single seed, using all $\binom{7}{2} = 21$ two-task models and all $\binom{7}{3} = 35$ three-task models, but only comparing non-overlapping pairs (models sharing no common tasks). This yields 105 non-overlapping pairs for 2-task models and 70 for 3-task models. Fig. 12(b) shows within-task CKA (same task combination, different seeds) as a function of task count, demonstrating that multi-task training also reduces seed-to-seed variability: representations become more consistent not just across tasks but also across random initializations.

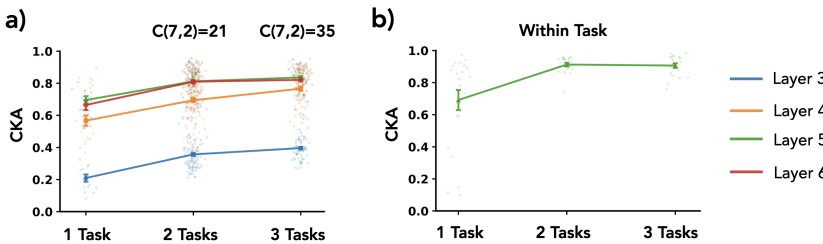

Figure 12: **Aggregated CKA analysis.** (a) CKA vs. task count for single seed, comparing only non-overlapping model pairs (105 pairs for 2-task, 70 pairs for 3-task). (b) Within-task CKA (same task combination, different seeds) increases with task count, indicating multi-task training reduces seed variability.

**CKA vs. Generalization (Annotated).** Fig. 13 is an annotated version of Fig. 3(b), with each point labeled by its (train$\rightarrow$eval) task pair.

972
973
974
975
976
977
978
979
980
981
982
983
984
985
986
987
988
989
990
991
992
993
994
995
996
997
998
999
1000
1001
1002
1003
1004
1005
1006
1007
1008
1009
1010
1011
1012
1013
1014
1015
1016
1017
1018
1019
1020
1021
1022
1023
1024
1025

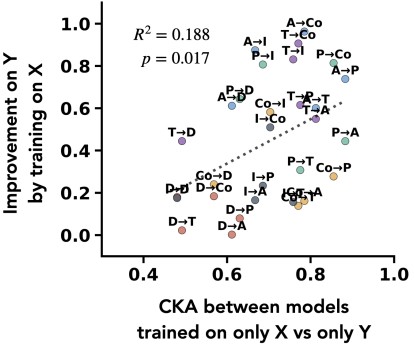

Figure 13: **Annotated version of Fig. 3(b).** Each point is labeled with its (train→eval) task pair. D=distance, T=triangle area, A=angle, Co=compass, I=inside, P=perimeter.

### E.4 Additional Fine-Tuning Evaluation Results

Raw fine-tuning results for individual seeds.

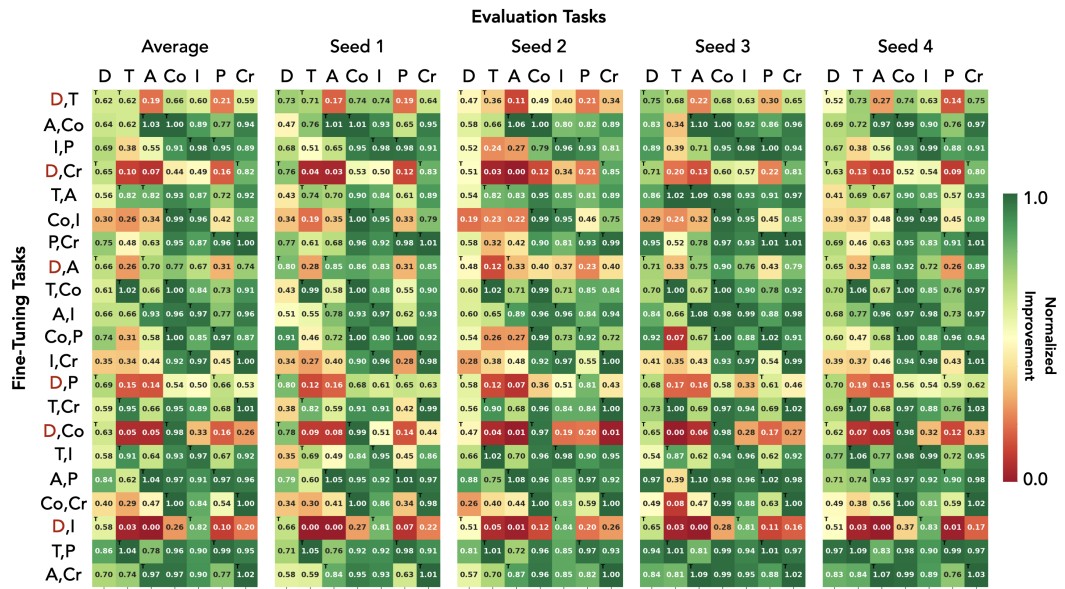

Figure 14: **Single-task fine-tuning results for individual seeds.** Per-seed version of Fig. 3(a), organized in a 2×2 grid.

Figure 15: **Two-task fine-tuning normalized improvement for all 21 task combinations.** Leftmost panel shows average across seeds; remaining panels show individual seeds.

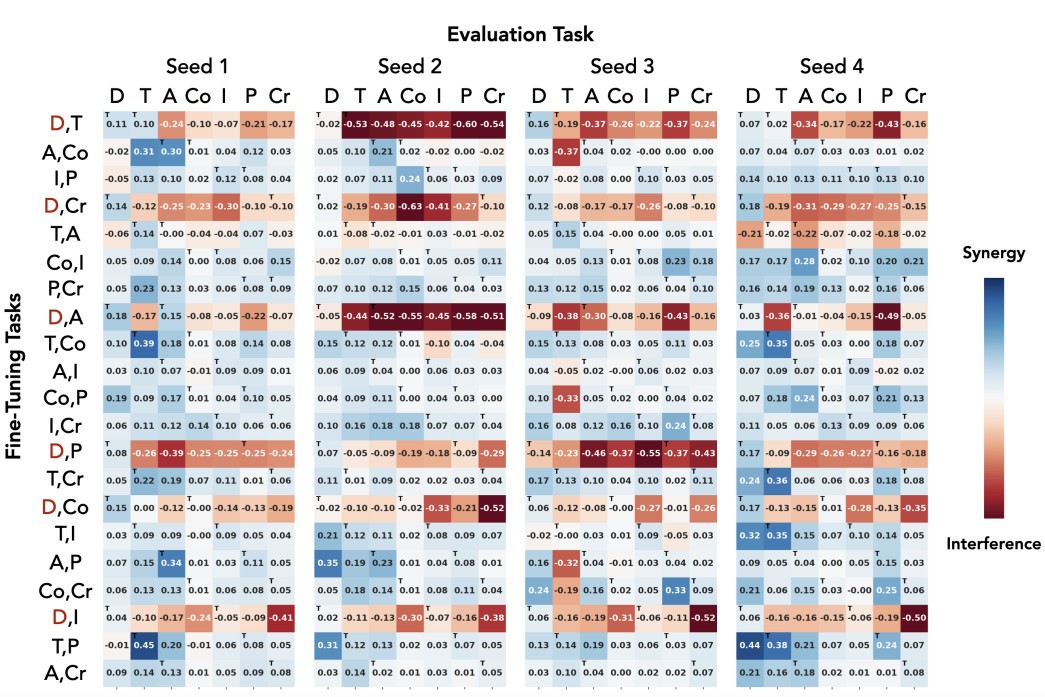

Figure 16: **Deviation from best-teacher expectation for all 21 two-task combinations.** All 4 seeds shown; average is in main text Fig. 4(c).

### E.5  PRETRAINING VARIATIONS

**Pretraining with `Atlantis`.**    In the main text, we showed that fine-tuning on divergent tasks fails to integrate `Atlantis` cities into the learned representation manifold (Fig. 4d, red histogram). To verify that this failure stems from fine-tuning dynamics rather than a peculiarity of the geometry around `Atlantis`, we trained a model with `Atlantis` cities included from the start of pretraining. Fig. 17 shows the resulting representations: `Atlantis` cities are seamlessly integrated into the world manifold, indistinguishable from other cities in both PCA projections (a) and linear probe reconstructions (b). This confirms that the representation space can readily accommodate `Atlantis`, and thus, the integration failure observed in fine-tuning is a property of the optimization dynamics, not a fundamental limitation of the architecture or task.

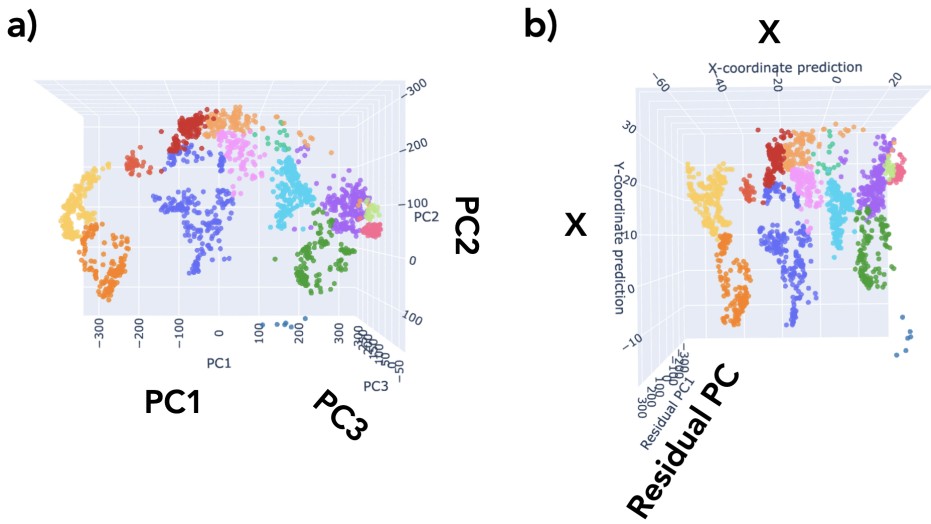

Figure 17: **Representations when `Atlantis` is included during pretraining.** (a) PCA projection showing `Atlantis` cities (small cluster in Atlantic region) integrated with world cities. (b) Linear probe reconstruction confirming geographic accuracy. Unlike fine-tuned models, `Atlantis` cities lie on the same manifold as other cities.

**Wider Model.**    To test whether our findings depend on model capacity, we trained a wider model with 2× the hidden dimension (256 vs. 128) and intermediate size (1024 vs. 512), resulting in approximately 4× the parameters. Fig. 18 shows fine-tuning results for this wider model: (a) single-task fine-tuning normalized improvement; (b) two-task fine-tuning normalized improvement; (c) deviation from best-teacher expectation. We still observe that `distance`-containing combinations (red labels in panel c) show degraded cross-task generalization. This suggests that divergent task interference is not simply a capacity limitation.

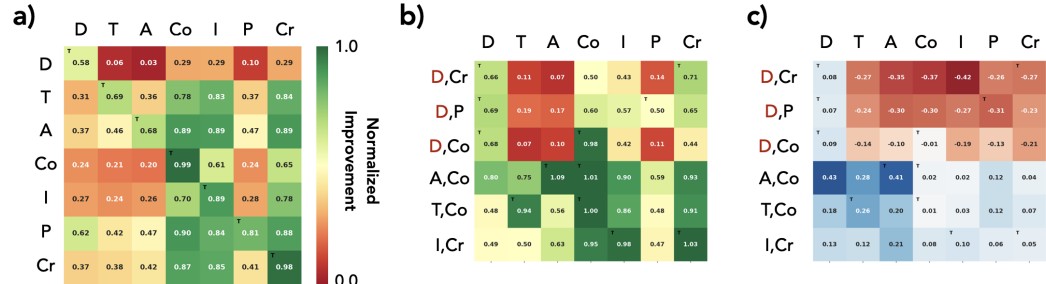

Figure 18: **Fine-tuning results for wider model (2× hidden dimension).** For all panels: rows = fine-tuning task(s), columns = evaluation task. (a) Single-task fine-tuning normalized improvement. (b) Two-task fine-tuning normalized improvement. (c) Deviation from best-teacher expectation; `distance`-containing combinations (red labels) still show degraded generalization.

