# OpenReview forum: "Divergent Tasks Harm Integration Of New Entities Via Fine-Tuning"
_ICLR.cc/2026/Workshop/Sci4DL — Sci4DL 2026_

### Official Review · Reviewer_pdBF · 2026-02-19

**Fit:** 2
**Significance:** 2
**Confidence:** 2

**Summary:**

This paper examines the effects of different fine-tuning tasks when attempting to add new data to a model. The paper examines how well fine-tuning on one task may affect the performance on another. For the case of fine-tuning on pairs of tasks, they show that there exists tasks which reduce the performance gains when included. By way of explaining the findings,  the paper examines the representation of the learned data and shows that the novel data may be stored in a different subspace, depending on the training tasks.

To do this, they train a transformer on several geometric tasks based on real-world city data. (Distances, Enclosed Area, Directions etc)
They then fine tune it on cities that are
* Synthetic
* Out of distribution (in the Atlantic ocean, far from real-world cities in the original training data)

When the "Distance" task is paired with other tasks for fine-tuning, the performance is degraded relative to that which can be achieved by fine-tuning on the other task only.
In an exemplar, fine-tuning on the pair of "Distance" and "Perimeter" tasks encodes new data in separate subspaces, rather than the shared manifold, whereas "Angle" + "Compass" appears to integrate well.
It is observed that the CKA between models trained only the single tasks correlates with cross-task generalisation (i.e. improvement on task Y when fine-tuning on task X is positively correlated with the alignment between models trained only on tasks X or Y). This observation excludes models that failed to train on the "Crossing" single-task.
It is claimed that the tasks that are "Divergent" (i.e. have a large CKA between models trained on different single-tasks) harm the integration of new data.

**Strengths:**

1) Reproducable effect: It is noteworthy that the two-task finetuning results in Fig 15 are similar between seeds so this is not a random or cherry picked phenomenon
2) Opportunity for further research: The corelation between single-task model CKA, and inter-task performance improvement is intriguing
3) It is an interesting and counter-intuitive feature that fine-tuning on an apparently related task can reduce the benefit of fine-tuning on another task
4) Potential novel contribution to training and evaluation: The representation-space linear probing to show how points are incorporated offers a potential way of understanding the poor fine-tuning and may indicate an approach to detecting improper integration of new information during fine-tuning

**Suggestions:**

1) Narrower, or more clearly justified, scope of claim: Might it be fairer to soften the title and claim to reflect the fact that this was only one task (Distance) on one architecture and this may not reflect a broad trend. Or perhaps the breadth of claim is valid but requires more explicit justification.
2) Clarification of main claim language: In the abstract (at least) the phrase ""actively harm performance on other tasks"" is used. To me, this implies that the performance goes down relative to pre-training performance, although I believe the finding is that the improvement is not as great as it was for single-task training. Is it possible to rephrase this for clarity?
3) Justification of cross-generalisation inverse relationship: Further analytical details could help lay clear the claim at Lines 119-125. Currently it is not clear to what extent the relationship is an inverse one. However, the broad observation that tasks are not necessarily mutually cross-generalisable is useful.
4) Justification of Task Divergence as a generally harmful feature: In Result 4 the conclusion is about ""Divergent Tasks"" but only seems to identify Distance as harming generalisation. Perhaps add an analysis similar to that of Figure 3(b), but for twin-task training? i.e. something to show that divergence is generally correlated with the twin-task training harm?
5) Expand on significance of out-of-distribution fine-tuning: Could a line be added to explain why out-of-distribution synthetic data was chosen? For example, do the observed effects not appear when fine-tuning on held-out, real-world cities, or synthetic cities on real land-masses?

---

### Official Review · Reviewer_gy7V · 2026-02-26

**Fit:** 2
**Significance:** 1
**Confidence:** 2

**Summary:**

The authors sought to understand the effects of fine tuning on task generalization. This manuscript has a nonlinear and unconventional format. The main body text does not provide the reader with sufficient background to put the work in context and confuses methods, results, and discussion sections.

**Strengths:**

- The manuscript is well-illustrated
- Authors communicate key findings in bolded text
- It's clear that prior work is well-surveyed, but only in the Appendix
- Briefly mentions limitations

**Suggestions:**

I recognize that the authors have a detailed appendix section to keep the main body of the manuscript to four pages, but authors have an obligation to summarize this information in the main body text so that it is a cohesive document.

- Distill related work and provide this high level overview so that readers understand where it sits relative to state of the art
- Similarly provide a high level overview of methods immediately following the introduction and related work
- Disaggregate methods, results, and discussion into separate subsections in the body text
- Define and cite on first reference key terms, such as CKA

---

### Official Review · Reviewer_qkYW · 2026-02-27

**Fit:** 2
**Significance:** 3
**Confidence:** 2

**Summary:**

The paper studies how the set of pretraining tasks affects cross-task generalisation when the model is fine-tuned to incorporate new entities. To this end, small transformers are trained on 7 geometric tasks defined over 5075 real-world city coordinates.  These geometric tasks include computing the distance, triarea, and so on. The authors find that the cross-task generalisation for their example strongly depends on the fine-tuning task. Moreover, this variation is predicted to a good extent by the representational similarity between the multi-task pretrained model and a model trained on a single task. The concept of divergent task is introduced, which is a task with low representational similarity. It is observed that pretraining with the divergent task included significantly decreases generalisation. Furthermore, the presence of divergent tasks in the pretraining task set also harms the integration of new entities for the fine-tuning stage.

**Strengths:**

The paper is written in a concise and clear manner. The questions that this work poses, the results, and the limitations are all explained well. Moreover, the questions that the authors are trying to answer are very relevant for designing the pretraining paradigm of a model, namely what tasks to include in this pretraining. The selected data set contains very well defined and interpretable tasks, conducive to providing answers that are not obfuscated by data idiosyncrasies. The methodology is sound and leads to a robust set of results.

**Suggestions:**

The authors could include a clear real-world example where it would be useful to conduct the type of analysis that they claim is useful in their work. One could argue that in most cases, one cannot just remove the divergent task from the pretraining. The addition of this information would contextualise the paper much better.

Secondly, after reading the paper there is one important question, in my opinion, that is left unanswered: what makes the distance a divergent task, aside from having a large CKA? This task shows lower overall performance. I would be cautious that this study is not just saying that if you remove the hard task from the pretraining, you get overall better performance. Hence, studying why distance behaves this way and perimeter does not could be worthwhile.

Finally, the figures are extremely hard to read. Some statements in the paper reference the figures and I had to take them for granted, since I could not verify them in the figure. The paper would benefit from removing some of its figures and decluttering the rest.

---

### Meta-Review · Area_Chair_RBmd · 2026-03-01

**Recommendation:** Accept

**Metareview:**

Nice contribution, great fit. Recommending acceptance.

---

### Decision · Program_Chairs · 2026-03-02

Accept